# Assessment of Different Niosome Formulations for Optogenetic Applications: Morphological and Electrophysiological Effects

**DOI:** 10.3390/pharmaceutics15071860

**Published:** 2023-07-01

**Authors:** José David Celdrán, Lawrence Humphreys, Desirée González, Cristina Soto-Sánchez, Gema Martínez-Navarrete, Iván Maldonado, Idoia Gallego, Ilia Villate-Beitia, Myriam Sainz-Ramos, Gustavo Puras, José Luis Pedraz, Eduardo Fernández

**Affiliations:** 1Biomedical Neuroengineering, Institute of Bioengineering (IB), University Miguel Hernández (UMH), 03020 Elche, Spain; jose.celdran.lopez@gmail.com (J.D.C.); lawrencehumphreys@hotmail.com (L.H.); dgonzalez@umh.es (D.G.); csoto@umh.es (C.S.-S.); gema.martinezn@umh.es (G.M.-N.); 2Networking Research Centre of Bioengineering, Biomaterials and Nanomedicine (CIBER-BBN), Carlos III Health Institute (ISCIII), 28029 Madrid, Spain; ivan.maldonado@ehu.eus (I.M.); idoia.gallego@ehu.eus (I.G.); aneilia.villate@ehu.eus (I.V.-B.); miriam.sainz@ehu.eus (M.S.-R.); joseluis.pedraz@ehu.eus (J.L.P.); 3Bioaraba, NanoBioCel Group, School of Pharmacy, University of the Basque Country (UPV/EHU), 01006 Vitoria-Gasteiz, Spain

**Keywords:** niosomes, optogenetics, morphological effects, electrophysiological effects

## Abstract

Gene therapy and optogenetics are becoming promising tools for treating several nervous system pathologies. Currently, most of these approaches use viral vectors to transport the genetic material inside the cells, but viruses present some potential risks, such as marked immunogenicity, insertional mutagenesis, and limited insert gene size. In this framework, non-viral nanoparticles, such as niosomes, are emerging as possible alternative tools to deliver genetic material, avoiding the aforementioned problems. To determine their suitability as vectors for optogenetic therapies in this work, we tested three different niosome formulations combined with three optogenetic plasmids in rat cortical neurons in vitro. All niosomes tested successfully expressed optogenetic channels, which were dependent on the ratio of niosome to plasmid, with higher concentrations yielding higher expression rates. However, we found changes in the dendritic morphology and electrophysiological properties of transfected cells, especially when we used higher concentrations of niosomes. Our results highlight the potential use of niosomes for optogenetic applications and suggest that special care must be taken to achieve an optimal balance of niosomes and nucleic acids to achieve the therapeutic effects envisioned by these technologies.

## 1. Introduction

Gene therapy aims to treat diseases by introducing genetic material (DNA or RNA) into the cells of the patients, either by correcting, adding, or removing a genetic sequence. It was first proposed in the 1970s for treating monogenic disorders [1] and has since been at the forefront of cancer treatment [2]. Gene therapy has achieved considerable success, with over two dozen official drugs approved for clinical trials [3,4]. One key component for delivering the genetic material into cells is the delivery system, classically referred to as a vector, of which there are many, each with its advantages and disadvantages. Among the most promising vectors that have proven to be suitable for genetic delivery are viral vectors [5,6,7]. They exhibit stable long-term expression and high transgene levels. Nevertheless, these vectors have some drawbacks, such as immunoreactivity, toxin production, insertional mutagenesis, and limitations in the size of genes that can be carried by the viral vector [6,7,8,9,10]. In this sense, non-viral nanoparticle vectors have been proposed as an interesting alternative that can overcome some of the aforementioned issues. More specifically, they have low immunogenicity and low cytotoxicity, are easily manufactured, and do not have the same gene-size restrictions [11]. In particular, niosomes have proven to be a promising candidate to deliver genetic material through the cell membrane [12,13,14].

Niosomes are bilayer vesicles composed of three main components—cationic lipids, “helper” components, and non-ionic surfactants—and can bind to DNA (forming nioplexes), exhibit long-term stability and proper physicochemical properties, and have relatively low preparation costs [15,16,17,18,19]. Our group already has extensive experience using niosomes as vectors for genetic material in the retina [19,20,21,22,23] and the brain cortex [24,25,26], achieving encouraging results in both. This opens the possibility of developing a safer and non-toxic genetic therapy for the treatment of multiple diseases that affect neural tissues. Other research groups have used niosomes for gene therapy, as well as for the delivery of genetic material into mesenchymal stem cells [27] and even in retinal tissue [28], but most of them have put their efforts into treating cancer by gene-silencing therapy with miRNAs [29], siRNAs [30,31], and oligodeoxynucleotides [32]. Though niosomes have many advantages, there is no scientific evidence related to their use for optogenetic applications.

Here, to the best of our knowledge, we are the first to combine niosome-based genetic delivery with optogenetic plasmids into cortical tissue. Optogenetics is a method that uses targeted ectopic expression of light-activated proteins (opsins) to control cell-specific neural activity with millisecond precision [33], allowing for precise activation of neural circuits using specific promoters [34,35]. Optogenetics has thus emerged as a promising alternative for treating diseases, such as epilepsy [36] and Parkinson’s [37] and even a current clinical trial in a blind patient [38]. Two improved optogenetic variants have emerged with distinct advantages for photostimulations: ChrimsonR [39] and CatCh [40]. ChrimsonR is a red-shifted channelrhopsin (activated at 590 nm) with fast-kinetics and high cellular trafficking, while CatCh is a blue-light-activated channelrhodopsin (activated at 470 nm) that introduces calcium into cells and possesses fast-kinetics. However, these optogenetic proteins have been delivered into neuronal cells using viral vectors or in utero electroporation [38,39,40], but not niosomes. Therefore, delivering these optogenetic tools into neuronal cells using niosomes as vectors (which do not have some of the disadvantages of the viral vectors) could be a promising approach to explore.

In this study, we delivered optogenetic plasmids that codify for ChrimsonR and CatCh, as well as the ubiquitously used GFP plasmid as a control [41], combined with three different niosome formulations into rat cortical neurons in vitro. These niosome formulations varied among them in the “helper” compound (nanodiamonds (ND12), sphingolipids (P10), and chloroquine (CQ)), whose capabilities of delivering reporting genetic material into cells of multiple tissues have been previously successfully tested [42,43,44]. Our main objective was to determine the suitability of this approach for the photosensitization of cortical neurons in optogenetic therapies.

In this work, we characterized the morphology, electrophysiology, and cell viability of different combinations of niosomes and optogenetic plasmids in rat cortical in vitro cultures.

## 2. Material and Methods

### 2.1. Elaboration of Niosomes and Nioplexes

All niosome formulations were elaborated by the oil-in-water emulsification technique [17]. In the case of ND12 niosomes, nanodiamonds (NDs) were purchased as ultrananocrystalline diamonds with grain sizes smaller than 10 nm (Sigma-Aldrich, Burlington, MA, USA). A volume of 250 µL of NDs (10 mg/mL in H_2_O) was ultrasonicated for 30 min and mixed with 2 mL of 0.5% Tween^®^ (Sigma-Aldrich, Madrid, Spain), and 1.75 mL of MilliQ^®^ water as the aqueous phase. A total of 5 mg of 1,2-di-O-octadecenyl-3-trimethylammonium propane chloride salt (DOTMA; Avanti Polar Lipids, Inc., Alabaster, AL, USA) was accurately weighted to obtain 1/2 ND/DOTMA mass ratios. The DOTMA was diluted in 1 mL of dichlorometane (DCM; Panreac, Castellar del Vallès, Barcelona, Spain), which constituted the organic phase. This phase was added to the aqueous phase and immediately sonicated for 30 min at 50 W (Branson Sonifier 250, Danbury). DCM was evaporated for 2 h at room temperature under magnetic stirring, obtaining a cationic lipid concentration of 1.2 mg/mL.

P10 niosomes were obtained by combining DOTMA with 2-[2-[(2R,3R)-3,4-bis(2-hydroxyethoxy)oxolan-2-yl]-2-(2-hydroxyethoxy)ethoxy]ethyl dodecanoate (Polysorbate 20, Bio-Rad, Alcobendas, Madrid, Spain) non-ionic surfactant and mixing them with sphingolipids from animal origin found in the intestinal mucosa of mammals with high levels of sphingomyelin (Bioiberica laboratory, Sus scrofa, pig) as helper components. Briefly, 3.4 mg of cationic lipid was gently grounded with 100 µg of sphingolipids, and then 500 µL of DCM was added to this lipid mixture and emulsified with 2.5 mL of polysorbate 20 (0.5%, *w*:*w*). Components were sonicated for 30 s at 50 W. Next, the DCM organic solvent was evaporated and eliminated from the emulsion by using a magnetic stirrer for 2 h at room temperature inside an extraction hood. Upon DCM evaporation, a colloidal dispersion carrying the formulations was obtained with a final cationic lipid concentration of 1.5 mg/mL.

CQ niosomes were prepared by dissolving 5 mg of cationic lipid 2,3-di(tetradecyloxy)propan-1-amine, 12.5 mg of non-ionic tensioactive poloxamer 407 (Sigma-Aldrich, Burlington, MA, USA), and 12.5 mg of non-ionic tensioactive polysorbate 80 (Sigma-Aldrich, USA) in 1 mL of DCM. The water phase contained 2.5 mg of “helper” lipid chloroquine diphosphate salt (Sigma-Aldrich, Burlington, MA, USA) dissolved in 5 mL of distilled water. The organic phase and the water phase were emulsified by sonication for 30 s at 50 W. The organic solvent was removed from the emulsion by evaporation under magnetic agitation for 3 h at room temperature, obtaining a cationic lipid concentration of 1 mg/mL.

Nioplexes were obtained by incubating each type of niosome with each plasmid. Optogenetic plasmids pCAG-ChrimsonR-tdT and pAAV-Syn-ChrimsonR-tdT were obtained from the Edward Boyden (Addgene plasmid # 59169 and Addgene plasmid # 59171, respectively), optogenetic plasmid pCMV-CatCh-EYFP was a gift from Peter Hegemann, and pCAG-GFP was obtained from Connie Cepko (Addgene plasmid # 11150). Plasmids were expanded and purified using the Qiagen endotoxin-free plasmid purification Maxi-prep kit (Qiagen, Valencia, CA, USA), according to the manufacturer’s instructions. The concentration of the purified plasmid was quantified in a NanoDrop™ 2000 Spectrophotometer (Thermofisher Scientific, Waltham, MA, USA). Then, an appropriate volume of each plasmid was mixed and incubated for 30 min at room temperature with the corresponding volume of each niosome suspension (1 mg/mL cationic lipid) to obtain the respective nioplexes at cationic lipid:DNA ratios (*w*:*w*) of 2:1, 5:1, 8:1, and 10:1.

### 2.2. Physicochemical Characterization of Niosomes and Nioplexes

The hydrodynamic diameter, which includes the particle size and dispersity (Ð), and the zeta potential of all niosomes and their corresponding nioplexes were determined by Dynamic Light Scattering (DLS) and by Laser Doppler Velocimetry (LDV), respectively, using a Zetasizer Nano ZS (Malvern Instrument, Malvern, Worcestershire, UK), and the morphology of niosomes was determined by transmission electron microscopy (TEM), all as previously described [22].

### 2.3. Animal Models

E16-E19 rat embryos (Sprague Dawley) were used for the extraction of primary neuronal cells for in vitro experiments. All experimental procedures were carried out in accordance with the RD 53/2013 Spanish and 2010/63/EU European Union regulations for the use of animals in scientific research. Procedures were approved and supervised by the Miguel Hernández University Standing Committee for Animal Use in the Laboratory with code UMH.CID.DPC.01.17.2019/VSC/PEA/0010.

### 2.4. Primary Neuronal Cell Extraction and Culture

Primary neuronal cells were extracted from the cortical tissue of rat embryos (Sprague Dawley) and maintained in DMEM (GIBCO^®^, Thermofisher Scientific, Waltham, MA, USA) with 10% fetal bovine serum (FBS; Biowest^®^, Nuaillé, Pays de la Loire, France) during extraction. Afterwards, we removed DMEM with 10% FBS and added FBS-free DMEM in order to perform chemical dissociation. Chemical dissociation was carried out by adding trypsin 0.05% and incubating the mixture at 37 °C. Once the cell density was quantified in a hemocytometer, cells were re-suspended according to the desired morphological analysis. For morphological analysis of cortical neurons transfected with nioplexes, cells were resuspended in Neurobasal™ (GIBCO^®^) medium supplemented with FBS, B27, GlutaMAX, and penicillin-streptomycin (GIBCO^®^) and seeded at 1.5 × 10^5^ cells per well in 24-well plates on glass coverslips, while for morphological analysis of cortical neurons transfected with only plasmids or niosomes, cells were resuspended in BrainPhys™ Imaging Optimized Medium (BPI; STEMCELL^®^, Saint-Égrève, Grenoble, France) supplemented with FBS, B27, GlutaMAX, and penicillin-streptomycin (GIBCO^®^) and seeded at 5 × 10^4^ cells per well in 24-well plates on glass coverslips. Cell cultures were maintained in an incubator at 37 °C and 5% CO_2_. This BPI medium was used in order to maintain alive the smaller number of cells per well (compared to Neurobasal™ cultures) and to facilitate cell imaging without the presence of genetically expressed fluorescent reporters. For cell viability analysis, cells were re-suspended in Neurobasal™ medium supplemented with FBS, B27, GlutaMAX, and penicillin-streptomycin and seeded at 1.5 × 10^4^ cells per well in 96-well plates.

### 2.5. In Vitro Transfection in Primary Neuronal Cell Culture

Cells in Neurobasal™ medium were seeded and incubated in 24-well plates between 21–28 days in vitro (DIV) before transfecting. Nioplexes, composed of niosomes and 1.25 µg of plasmid per well at their respective cationic lipid:DNA ratio (*w*:*w*), were formed by electrostatic interactions during 30 min at room temperature in serum-free OptiMEM solution (GIBCO^®^). Cells were transfected with nioplexes between these ages and not in younger cells (7–11 DIV) since they developed electrophysiological activity, which allowed correlating morphological results with electrophysiological results, also avoiding the loss of fluorescent cells while waiting for younger cells to develop electrophysiological activity. Transfection was carried out by exposing cells to nioplexes for 4 h at 37 °C in the incubator, followed by removal of the transfection medium and replacement with fresh Neurobasal™ medium. Lipofectamine™ 2000 (Invitrogen, Carlsbad, CA, USA) at a 1:1 ratio was employed as a positive control. In 96-well plates, the seeding, incubation, and transfection protocol was similar, but this time we also treated cells between 7–11 DIV, and the concentration of plasmid in each well was 0.25 µg.

Cells in BPI medium were seeded and incubated in 24-well plates, both between 21–28 and 7–11 DIV before transfecting. In addition, 1.25 µg of plasmid and niosomes at their respective cationic lipid/DNA ratio were incubated separately for 30 min at room temperature in OptiMEM solution. Transfection was carried out by exposing cells to either plasmid or niosomes for 4 h at 37 °C in the incubator, followed by removal of the transfection medium and replacement with fresh BPI medium. Untreated cells incubated with OptiMEM for 4 h were used as a positive control.

### 2.6. Morphological Evaluation of Transfected Cultured Cortical Neurons

Neurobasal™ medium was removed from cells 24 h after their exposure to nioplexes, and they were fixed by 4% paraformaldehyde (PFA; Sigma-Aldrich, Burlington, MA, USA) for 20 min and washed 2 times with phosphate buffer (PB; Sigma-Aldrich, Burlington, MA, USA) concentrated at 0.1 M, 10 min each time. After washing, PB 0.1 M with 0.5% Triton X-100 was added for 5 min, and the cell nuclei were stained with Hoechst 33342 (Sigma-Aldrich, Burlington, MA, USA) for 10 min. Coverslips were mounted in slides with Mowiol^®^ 4–88 (Sigma-Aldrich, Spain). Fluorescence images were taken with laser-confocal microscopy (Leica TCS SPE Microsystems GmbH, Wetzlar, Germany).

BPI medium was removed from cells 24 h after their exposure to either plasmids or niosomes, and they were fixed as described above and blocked with 10% FBS for 1 h, and then incubated overnight at 4 °C with anti-rabbit MAP2 monoclonal antibody (1 mg/mL, 1:500 dilution, Millipore, Burlington, MA, USA). After washing 2 times with PB 0.1 M for 10 min each time, cells were incubated for 1 h with a secondary antibody, AlexaFluor^®^ 488 donkey anti-rabbit (2 mg/mL, 1:1000 dilution, Invitrogen, Themofisher Scientific, Waltham, MA, USA). After 3 washes with PB of 5 min each wash, 0.5% Triton X-100 and Hoechst 33342 addition, as well as coverslip mounting and fluorescence image taking, were carried out as described above.

Morphological analysis of fluorescence images was performed by the Fiji plugin NeuronJ. The morphological parameters evaluated in rat cortical neurons were number of dendrites, branching points, total length of all dendrites, mean length of all dendrites, and longest dendrite.

### 2.7. Electrophysiological Recordings

Transfected coverslips were removed from wells using tweezers when the cells had 21–28 DIV and were kept in extracellular medium containing (in mM): 136 NaCl, 2.5 KCl, 10 HEPES, 10 Glucose, 2 CaCl_2_, 1.3 MgCl (pH = 7.3).

Borosilicate glass capillaries (1B150F-4, World Precision Instruments, USA) of 3–5 MΩ, obtained by a P97 puller (Sutter Instrument Co., Novato, CA, USA) for patch clamp recordings, were filled with an intracellular medium containing (in mM): 130 K^+^-gluconate, 10 NaCl, 1 EGTA, 0.133 CaCl_2_, 2 MgCl_2_, 10 HEPES, 3.5 MgATP, 1 NaGTP (pH = 7.3). Cells were targeted with a patch electrode under visual guidance using the reporter tag’s fluorescence, and whole-cell recordings were obtained using the HEKA EPC 10 USB double patch clamp amplifier (Harvard Bioscience, Inc., Holliston, MA, USA). Photocurrents were recorded while voltage-clamping cells at a potential of −60 mV, and also in the current clamp configuration in order to monitor the membrane potential during light stimulations.

A monochromatic light source (pE-300 ultra, CoolLED, Andover, UK) was used to stimulate cells during electrophysiological recordings. In order to measure the photostimulation of targeted cells, we used two 5 ms light flashes at 20 W, with a 1 s interspace of 590 nm for ChrimsonR positive cells and 470 nm for CatCh positive cells.

### 2.8. Cell Viability

The cell viability of primary neuronal cell cultures of both 21–28 and 7–11 DIV after their exposure to either plasmids or niosomes were analyzed 24, 48, and 72 h post-transfection by the tetrazolium salt 3-[4,5-dimethylthiazol-2-yl]-2,5-diphenyl tetrazolium bromide (MTT; Sigma-Aldrich, Burlington, MA, USA) colorimetric assay, and the absorbance was read at 570 nm in a 2100-C microplate reader (Neuvar Inc., Palo Alto, CA, USA), according to the manufacturer’s instructions. Untreated cells incubated with OptiMEM were used as positive controls, and data were normalized relative to these positive controls.

A similar procedure was performed with primary neuronal cell cultures 21–28 DIV, which were treated with nioplexes and analyzed 24 h post-transfection.

### 2.9. Statistical Analysis

Differences between two groups were evaluated using a Mann–Whitney U test in non-parametric conditions, whereas for multiple comparison, it was either performed using a one-way ANOVA or multiple *t*-tests. Data are expressed as mean ± SD. A *p*-value < 0.05 was considered statistically significant. Analyses were performed with the GraphPad Prism 8.0 statistical package.

## 3. Results

### 3.1. Nioplexes Transfection and Neuron Morphology

We tested niosome formulations for their capacity to transfect optogenetic plasmids, as well as the widely used reporter GFP plasmid. More specifically, cortical neurons were transfected with the niosomes ND12, P10, and CQ, which were complexed with different plasmids (pCAG-ChrimsonR-tdT, pAAV-Syn-ChrimsonR-tdT, pCMV-CatCh-EYFP, and pCAG-GFP) at different cationic lipid/genetic material ratios (2:1, 5:1, 8:1, and 10:1) after 21 and 28 days in vitro (DIV). We did not observe the expression for any plasmid or niosome combination at a ratio of 2:1 after 24 h. However, with higher ratios (5:1, 8:1, and 10:1), we observed cell expression after 24 h. pCAG-ChrimsonR-tdT and pCAG-GFP plasmids expression was observed using a ratio of 5:1 and higher, pAAV-Syn-ChrimsonR-tdT plasmid expression was observed using a ratio of 8:1 and higher proportions, and pCMV-CatCh-EYFP plasmid expression was only observed using a ratio of 10:1.

All niosomes were able to deliver the optogenetic plasmids and successfully transfect neurons. However, the morphology of nioplex transfected cells appeared somehow different in comparison to cells treated with the commercially available reagent Lipofectamine. To quantify and characterize the differences, we measured (1) the number of dendrites, (2) branching points, (3) total length of all dendrites, (4) mean length of all dendrites, and (5) longest dendrite. Our results showed that neurons treated with niosome formulations and CAG-ChrimsonR plasmid had a significant reduction of all parameters in comparison with neurons transfected with Lipofectamine (Figure 1A,B and Appendix A). Similar results were observed in neurons treated with the Syn-ChrimsonR plasmid (Appendix A). This trend was also observed in neurons treated with niosome GFP complexes (Figure 1C,D and Appendix A), except for the treatment with ND12 at 5:1, which showed no statistical significance in the mean length of all dendrites (*p*-value = 0.1653) (Appendix A).

As previously mentioned, CatCh niosome complexes were only expressed at a ratio of 10:1. In general, we observed a similar trend in terms of changes to morphology. However, for cells transfected with P10 at a ratio of 10:1, we observed no statistical difference when we measured for the longest dendrite compared to controls (*p*-value = 0.7180). Additionally, all niosomes complexed with the CatCh plasmid did not exhibit any differences in the mean length of all the dendrites (Appendix A). All *p*-values can be found in Appendix A.

### 3.2. Electrophysiological Properties of Transfected Neurons

Having observed some morphological changes in neurons treated with niosome complexes, we set out to test if this also translated into any electrophysiological changes. We performed patch clamp electrophysiology on cells transfected after 21–28 DIV with our niosome complexes. Neurons transfected with optogenetic nioplexes or with Lipofectamine were photostimulated with 2 pulses (5 ms duration and 1 sec inter-pulse). In general, Lipofectamine-treated cells (n = 7) showed robust inward currents and light-driven action potentials (Figure 2A). When we performed voltage clamp (VC) recordings on cells treated with P10/CAG-ChrimsonR complexes (Figure 2B), in general, we observed similar rise times when compared to Lipofectamine controls, except for P10 at 8:1 (pulse 1 *p*-value = 0.0255 *, pulse 2 *p*-value = 0.0024 *, n = 6), which was slower (Figure 2C). Peak amplitudes values for all ratios were significantly smaller in comparison to controls (pulse 1 P10 vs. lipo, 10:1 *p*-value = 0.01 *, 8:1 *p*-value = 0.0006 ***, 5:1 *p*-value = 0.0005 ***, pulse 2, 10:1 *p*-value = 0.0119 * n = 6, 8:1 *p* = 0.0006 ***, n = 6, 5:1 *p* = 0.0006 ***, n = 12) (Figure 2D). When we performed current clamp (CC) recordings, we observed significant longer rise times only at a ratio of 8:1, but not with 10:1 or 5:1 (pulse 1 P10 vs. lipo, 10:1 *p*-value ≥ 0.9999, 8:1 *p*-value = 0.0141 *, 5:1 *p*-value = 0.8234, pulse 2, 10:1 *p*-value ≥ 0.9999 n = 6, 8:1 *p*-value = 0.0083 ***, n = 6, 5:1 *p*-value = 0.5691 n = 7, lipo n = 7) (Figure 2E). Peak amplitudes were significantly smaller for all ratios (pulse 1 P10 vs. lipo, 10:1 *p*-value = 0.0013 **, 8:1 *p*-value = 0.0024 **, 5:1 *p*-value = 0.0017 **, pulse 2, 10.1 *p*-value = 0.0011 **, n = 6, 8:1 *p*-value = 0.002** n = 6, 5:1 *p*-value = 0.0016 **, n = 7) (Figure 2F). Overall, the rise times between Lipofectamine-treated cells and P10 were not significantly different, except for cells treated with a ratio of 8:1. However, considerable electrophysiological changes were observed in terms of the peak amplitude when cells were photostimulated. This general trend was observed with cells treated with ND12 and CQ (Appendix A).

Neurons transfected with Syn-ChrimsonR plasmid and niosomes ND12 and P10 generally present statistical significance in rise time compared to Lipofectamine (Appendix A), except in the ND12 (10:1) treatment in both pulses (*p*-value = 0.3775 and 0.0505, respectively, in VC recordings) (Appendix A) and the P10 (10:1) treatment in pulse 1 (*p*-value = 0.2445 in VC) (Appendix A). CQ-Syn-ChrimsonR-treated cells showed no significant differences in rise times, except for the first pulse (*p*-value = 0.019 *) when using a ratio of 10:1 in CC recordings (Appendix A).

Peak amplitude values in both VC and CC modes showed statistical significance, except the ND12 (10:1) treatment in both pulses (*p*-value = 0.5061 and 0.6587, respectively) (Appendix A), P10 (8:1) treatment in both pulses (*p*-value = 0.5944 and 0.6644, respectively), P10 (10:1) treatment in both pulses (*p*-value = 0.3274 and 0.3756, respectively) (Appendix A), and CQ (10:1) treatment in pulses 1 and 2 (*p*-value = 0.4276 and 0.4907, respectively) (Appendix A).

Using the CatCh plasmid, there was no difference in any niosome treatment with any parameter (Appendix A), except the peak amplitude in VC between Lipofectamine and the P10 (10:1) treatment in pulse 1 (*p*-value = 0.0349 *) (Appendix A).

Overall, niosome treatment results in variable changes to the rise time in light-driven responses either decreasing or increasing, depending on the niosome formulation. Peak amplitudes exhibited similar trends in that most treatments resulted in reduced peak amplitudes in both the VC and CC modes compared with Lipofectamine. Although cells depolarized upon illumination in almost all cases, they never reached a threshold to fire an action potential.

To establish whether there was a positive correlation between morphology and electrophysiology, we compared the total length of all dendrites and the peak amplitude for both the VC and CC modes. All niosomes complexed with CAG-ChrimsonR showed a strong positive correlation in both the VC and CC modes with pulses 1 and 2 (pulse 1, VC R squared = 0.9442, CC R squared = 0.9327, pulse 2, VC R squared = 0.9432, CC R squared = 0.9219) when compared to the Lipofectamine control (Figure 3A–D). Syn-ChrimsonR treatments showed a positive correlation and statistical significance between morphology and electrophysiology in the VC mode peak amplitude (Appendix A), but the CC mode peak amplitude showed no statistical significance and a weaker positive correlation (Appendix A). CatCh plasmid treatments showed no statistical significance and a weak positive correlation in the both VC and CC modes with both pulses (Appendix A).

The strong correlation we observe using the CAG-ChrimsonR nioplex confirms that the morphological changes have a direct influence on the electrophysiological behavior of these cells. This was observed to a lesser extent with Syn-ChrimsonR only in the VC mode, while cells treated with CatCh exhibited a weak correlation for both pulses in both recording modes.

### 3.3. Morphological Characterization and Cell Viability Assesment of Niosomes, Nioplexes, and Naked Plasmids

To discern whether the niosomes themselves were the negative factor affecting morphology, rather than a combination with the plasmids, we treated cortical neurons after 21–28 DIV with only the niosome formulations and characterized the morphology by marking cells with the anti-MAP2 antibody. We observed significant differences in the number of dendrites, as well as in the total length of all dendrites, for several niosomes treatments when compared to untreated cells (Figure 4A,B), except for the total length of all dendrites of neurons treated with CQ (5:1) (*p*-value = 0.1051). Furthermore, CQ (5:1)-treated cells had a similar appearance to untreated controls, and dendritic blebbing was absent (Figure 4C,D) which was observed in most other treatments (Figure 2E). On the other hand, in some of the treatments, no statistical differences were observed in certain morphological parameters, more specifically, in the mean length of all dendrites with treatments ND12 (8:1) (*p*-value = 0.0753), ND12 (10:1) (*p*-value = 0.0630), P10 (5:1) (*p*-value = 0.1431), P10 (8:1) (*p*-value = 0.0630), and CQ (10:1) (*p*-value = 0.1655) (Appendix A).

To rule out the possibility that the plasmids themselves may have an adverse effect on neurons, we performed similar experiments by applying our naked plasmids, without niosome delivery systems, in 21–28 DIV cortical cultures. In general, no statistical differences were observed in the mean length of all dendrites and the longest dendrite in comparison to untreated controls (Appendix A). However, when using the CatCh and Syn-ChrimsonR plasmid, we did observe a reduction in the number of dendrites (*p*-value = 0.0136 * and 0.0014 **, respectively), branching points (*p*-value = 0.0081 ** and 0.0023 **, respectively), and total length of all dendrites (*p*-value = 0.0068 ** and 0.0039 **, respectively), and we also observed a reduction in the number of dendrites (*p*-value = 0.0498) and branching points (*p*-value = 0.0410) with the CAG-ChrimsonR plasmid, but not in the total length of all dendrites (*p*-value = 0.2799) (Appendix A). Experiments performed adding only Lipofectamine (1:1) showed no statistical difference in any morphological parameter compared with untreated neurons. All *p*-values can be found in Appendix A.

To investigate whether niosomes can affect the viability of neurons, we performed cell viability experiments. Neurons were seeded in a 96-well plate and divided into 3 groups: plasmid treated, niosome treated, and untreated controls. MTT was added to the wells 24, 48, and 72 h after the initial treatment. Values were normalized according to the absorbance values obtained in the untreated wells (Figure 5). The treatment with all nanoparticles at a 5:1 ratio did not show a statistically significant decrease in cell viability. On the other hand, wells treated with only nanoparticles at the 8:1 and 10:1 ratios showed a decrease in cell viability as early as 24 h, especially with the ND12 and CQ treatments (*p*-value < 0.0001 ****) and P10 to a lower extent (8:1 *p*-value = 0.03 *, 10:1 *p*-value = 0.038 *).

After 48 h of treatment, we only observed a significant decrease in cell viability in ND12 and P10 at 10:1 (*p*-value = 0.0367 * and 0.0198 *, respectively), but not at 8:1. However, CQ treatment at 8:1 and 10:1 resulted in a significant decrease in cell viability (*p*-value = 0.0001 **** and 0.0011 **, respectively). After 72 h, only the ND12 treatment at a 10:1 ratio showed a significant decrease in cell viability (*p*-value = 0.0251 *). These results suggest that the majority of the deleterious effects produced by these niosomes occurs within the first 24 h once the cells have initially internalized them. All *p*-values can be found in Appendix A.

We then repeated these experiments with the niosomes complexed with the plasmids to assess any combinatorial effect on cell viability. Overall, we observed a more significant decrease in cell viability in comparison to the niosome treatment by itself. At ratios 8:1 and 10:1 with optogenetic plasmids, we observed significant decreases in cell viability for all nioplexes tested (Figure 6), except for P10 at 8:1 for all plasmids and P10 at 10:1 in the CatCh plasmid (*p*-value = 0.1531). GFP nioplexes also resulted in decreased viability for all ratios and niosome combinations, except for P10 8:1 (*p*-value = 0.1807), CQ 8:1 (*p*-value = 0.0663), and P10 10:1 (*p*-value = 0.7845) (Figure 6). When we only applied the plasmids or Lipofectamine, we observed no decrease in cell viability (Appendix A). All *p*-values can be found in Appendix A.

To assess if these morphological changes could be age-dependent, we tested the niosomes on neurons cultured between 7 and 11 DIV. We observed robust statistical differences with all niosomes in the number of dendrites, total length of all dendrites (Figure 7), branching points, and longest dendrite (Appendix A) under all conditions. However, no statistical differences were found in any treatment in the mean length of all dendrites and some treatments for the longest dendrite (Appendix A). When we applied naked plasmids to younger cultures, no statistical differences were noted (Appendix A), except for the CAG-ChrimsonR plasmid in the total length of all dendrites (*p*-value = 0.0356 *) (Appendix A) and the naked GFP plasmid in the longest dendrite (*p*-value = 0.0199 *) (Appendix A) parameters. Additional experiments with only Lipofectamine (1:1) showed no statistical difference between untreated neurons in any morphological parameters, except in the branching points (*p*-value = 0.0122 *) and mean length of all dendrites (*p*-value = 0.0054 **) (Appendix A). All *p*-values can be found in Appendix A. Further comparisons of cell viability at different ages can be found in Figure 8 and Appendix A. All *p*-values can be found in Appendix A. These results suggest that older neurons appear to be more resistant to niosome-induced alterations.

## 4. Discussion

We have combined, for the first time, three different niosome formulations with three optogenetic plasmids and compared the transfections to a control GFP plasmid. Niosome ratios of 5:1, 8:1, and 10:1 demonstrated the capacity to transfect optogenetic material in rat cortical cells in vitro. However, a ratio of 2:1 proved to be insufficient with all plasmids tested. This is most likely due to the low concentration of niosomes and, in some cases, more negative zeta potential values hindering the electrostatic interaction between DNA and nanoparticles (Appendix A). Positively charged nioplexes are less likely to be formed that can easily interact with the negatively charged cell surface facilitating their uptake through endocytosis [22]. Although expression of the reporter confirmed the insertion of the optogenetic channels, we observed changes to the morphology of the neurons. These included a decrease in dendritic length and number, as well as arborization. This may be due to possible cytotoxic effects with the accumulation of high levels of cationic lipids and their headgroups (especially at the 8:1 and 10:1 proportions) [45]. When we used a ratio of 5:1, which has previously been reported to be optimal for transfection, here, once again, we observed a similar phenotype [22,42,44]. Some of the plasmids used required a minimal of an 8:1 ratio to observe expression (Syn-ChrimsonR). In some parameters, however, we did not observe any changes when compared to Lipofectamine-treated cells and niosomes, for example, with GFP and ND12 at a ratio of 5:1 in the mean length of all dendrites. This suggests that the size of the plasmid may also play some additional role, as the CAG-ChrimsonR plasmid is 7301 bp and the Syn-ChrimsonR plasmid is 6864 bp, while the GFP plasmid is 5551 bp.

Most of the literature on niosomes involves some form of stem cell or cancer cell line [27,29,30,31,32], which are generally more resistant to biological stresses in comparison to the more susceptible neurons, which may explain the effects we observed, especially at higher concentrations. However, previous studies on cell viability using niosomes on hepatocellular carcinoma, HepG2 cells, have shown that a 1.5 µM concentration results in 90% viability and 5 µM in less than 60% viability, corroborating the present results [46].

Interestingly, the CatCh plasmid would only express using a high ratio of 10:1; however, it showed no statistical difference in the mean length of all dendrites, even though the total number of dendrites and ramifications were reduced in comparison to Lipofectamine-treated cells. Considering that the CatCh plasmid is 7099 bp and, therefore, smaller than the CAG-ChrimsonR plasmid, the morphological alterations might be operating, at the molecular level, in a different way due to the size. One possible solution for solving problems related to plasmid size could be using lipid–polymer hybrid nanoparticles, which consist of nucleic acids polymers coated with a single lipid layer, allowing high nucleic acid condensation efficiency [47]. Others have reported improved efficiency of niosome drug delivery by magnetizing them and then modifying the surface by PEGylation to treat breast cancer cells with Carboplatin [48].

These morphological changes also translated into electrophysiological alterations. The nioplexes-treated neurons exhibit both reduced inward currents and depolarization, and generally were unable to fire when photostimulated, in contrast to Lipofectamine-treated neurons. During photostimulation, we did observe depolarization of the cells, but in almost all cases, these were sub-threshold and were unable to elicit any spikes. This suggests that the channels are transported to the cell membrane and are open during photoactivation; however, the reduction in the dendritic parameters is the most likely reason for the altered electrophysiological properties of the reduced photocurrents.

Peak amplitudes for all photocurrents were generally significantly reduced for all nioplexes when combined with CAG-ChrimsonR or Syn-ChrimsonR, while rise times depended on the plasmid and at which ratio. The CatCh plasmid, however, which was only expressed with all niosomes at a ratio of 10:1, exhibited in general slightly smaller peak amplitudes, but in most cases, they were not significantly different when compared to Lipofectamine. Rise times to peak amplitude were more varied and depended on the ratio and the nanoparticle being used. Neurons treated with the CatCh nioplexes had no differences in rise time compared with the CatCh with Lipofectamine.

We observed a strong correlation between the total length of all dendrites and peak amplitude between both pulses, suggesting that the morphological changes directly influence the cell’s electrophysiological properties. CAG-ChrimsonR revealed a strong positive correlation between morphology and electrophysiology. The noxious effects that niosome uptake has on normal dendritic morphology development directly relates to the lower peak values recorded. However, this correlation was not always consistent, being weaker in the CC peak amplitude of Syn-ChrimsonR expressing neurons and, in general, in CatCh expressing neurons. The CatCh results are consistent with the lower differences in both the morphological and electrophysiological values observed after treatments. Therefore, the difficulty of nioplex-treated neurons to fire APs could be related to the amount of cationic lipid introduced in the membrane, compromising stability [49].

To discern whether niosomes or plasmids were damaging the cells, we tested just the niosomes and the naked plasmids and characterized the morphology of neurons in young and mature cultures. To our surprise, both the niosomes and the naked plasmids appeared to have a detrimental influence on some morphological properties of the cells. This suggests that there may be a combinatory effect. Interestingly, while naked plasmids did not generally affect the morphology of younger neurons, they did affect the mature cultures in some aspects, especially with the Syn-ChrimsonR and CatCh plasmids. As the purity of the plasmids was within an acceptable range of (A260/A280 = 1.8–2), we doubt this may have been due to contamination. This was intriguing, as bigger plasmids tend to degrade faster within cell medium [50]. Further, while adult neurons seem not to be affected by Lipofectamine alone, in younger neurons, it affects some morphological parameters (especially the mean length of all dendrites), suggesting that young neurons are more sensitive also to Lipofectamine.

The nanoparticles are predominantly taken up into the cell through the process of endocytosis and initially trapped within a membrane vesicle, which budded off from the cell membrane and eventually released into the cytosol. Once the plasmid has been released from the vesicle, the nanoparticles may continue to disrupt intracellular membranes, such as those of the mitochondria. Mitochondrial metabolism that has been disrupted can lead to the generation of reactive oxygen species (ROS), which can damage DNA and exhibit other toxic effects. Some suggestions have been made that lipids that act as mRNA carriers in vaccines could be responsible for possible cytotoxicity in a similar fashion [51].

Finally, the viability results revealed that high proportions (specially, 8:1 and 10:1) of only niosomes produced a decline in cell viability, which was not observed in either naked plasmids or niosomes at a ratio of 5:1. This decline was consistent in young neurons throughout all times tested, but not in adult neurons, in which the noxious effect of niosomes appears to diminish with time. This is probably due to younger cultures being more sensitive to environmental changes in comparison to mature cells [52]. The results of cell viability with nioplexes in adult neurons suggested that nioplexes caused, in some cases, higher cell damage than only niosomes, as a higher decrease in cell viability was observed. Therefore, it seemed as if a synergistic effect between the niosomes and the plasmid occurs. This, however, varied among niosomes, with P10 nioplexes exhibiting the lowest decreases in cell viability, suggesting that the composition of the helper compound may have a key role in the degree of harm produced.

Our results show for the first time that all the niosomes tested were able to transfect cortical neurons with optogenetic channels, suggesting that niosomes are good candidates to transfect optogenetic tools. However, our results also suggest that there is room for improvement. Thus, some transfected cells exhibit morphological and electrophysiological changes that affect them negatively, especially when using high niosome concentrations. Therefore, we should be aware of the importance of finding a correct balance between the concentration of niosomes and the ratio of niosomes to optogenetic plasmids. Engineering new formulations is a critical challenge for the development of advanced optogenetics applications based on these promising non-viral vectors.

## Figures and Tables

**Figure 1 pharmaceutics-15-01860-f001:**
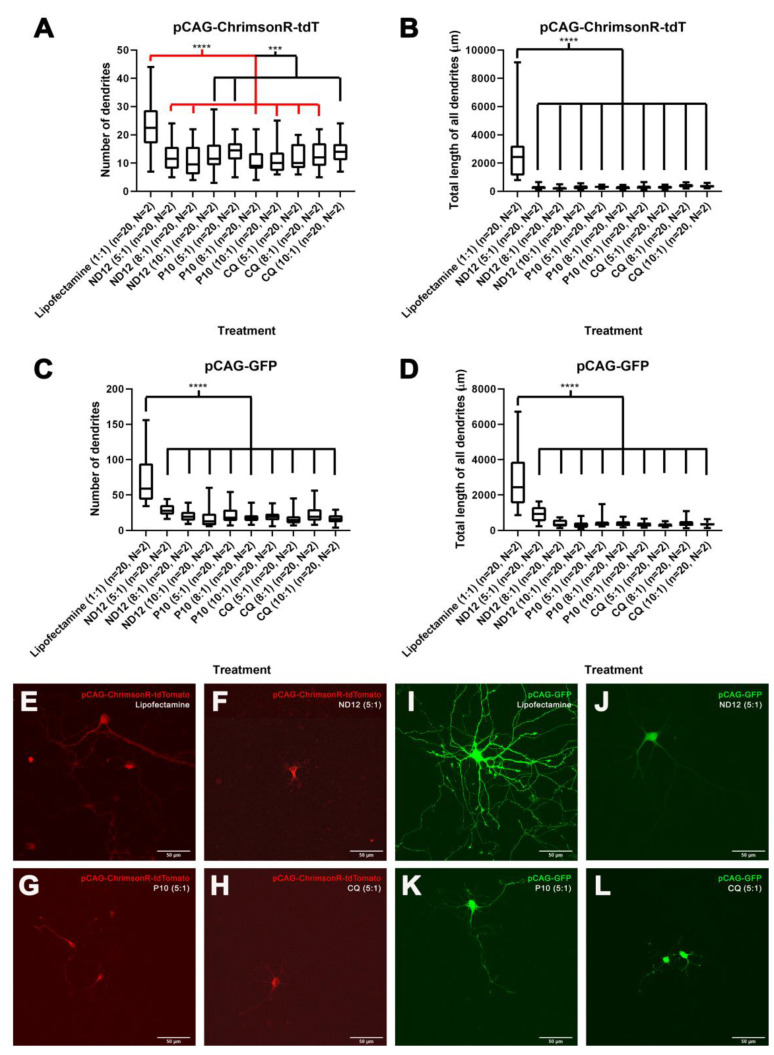
**Morphological changes induced by different niosome-based formulations.** The 21–28 DIV rat cortical neurons treated with nioplexes made of pCAG-ChrimsonR-tdTomato plus niosomes showed reduction in morphological parameters as number of dendrites (**A**) and total length of all dendrites (**B**) compared with the lipofectamine treatment, with the same outcome in pCAG-GFP (**C**,**D**) (Mann–Whitney test, *** *p* < 0.001, **** *p* < 0.0001, n = number of cells, N = number of cultures). (**E**–**H**) Morphological aspect of cortical neurons treated with pCAG-ChrimsonR-TdTomato plus lipofectamine (**E**), ND12(5:1) (**F**), P10(5:1) (**G**), and CQ(5:1) (**H**). (**I**–**L**) Morphological aspect cortical of neurons treated with pCAG-GFP plus lipofectamine (**I**), ND12(5:1) (**J**), P10(5:1) (**K**), and CQ(5:1) (scale bar = 50 µm) (**L**).

**Figure 2 pharmaceutics-15-01860-f002:**
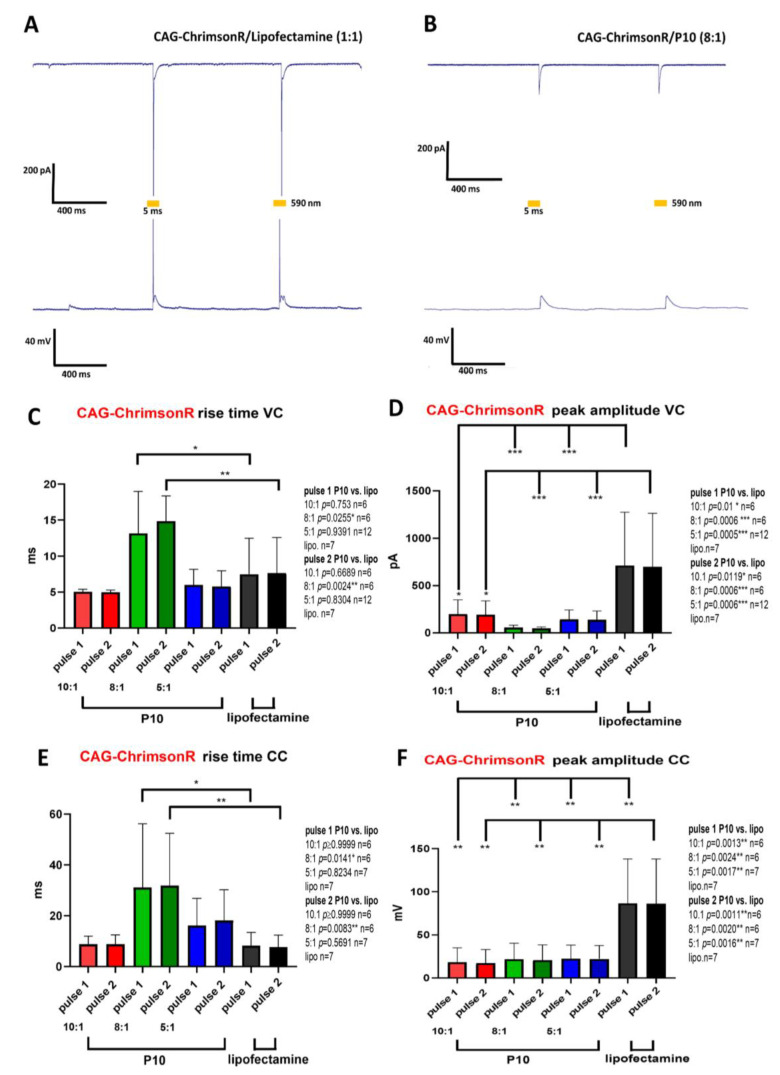
**Electrophysiological changes induced by different niosome-based formulations.** (**A**) Photostimulation of rat cortical neuron DIV 28 expressing CAG-ChrimsonR transfected with lipofectamine (1:1), showing AP firing. (**B**) Photostimulation of a rat cortical neuron DIV28 transfected with P10 (8:1) niosomes, showing depolarization, but no AP firing. Voltage clamp (VC, top) and current clamp (CC, bottom) recordings were performed while cells were photostimulated with 2 pulses of 5 ms (590 nm) with a 1-s interspace. (**C**–**F**) Comparison between lipofectamine and P10 niosomes at different ratios in rise time and peak amplitude electrophysiological parameters in each light pulse (ordinary one-way ANOVA, * *p* < 0.05, ** *p* < 0.01, *** *p* < 0.001). Graphs bars are expressed as mean ± SD.

**Figure 3 pharmaceutics-15-01860-f003:**
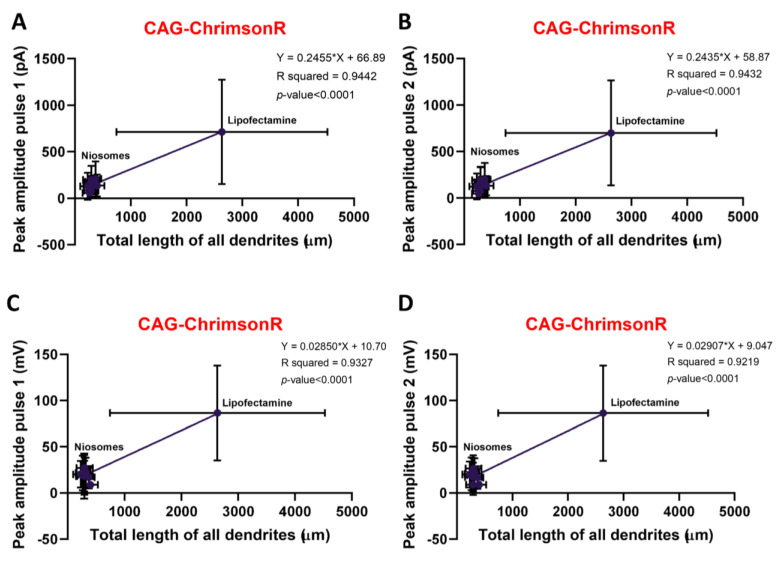
**Correlation of morphological and electrophysiological changes.** (**A**–**D**) Dispersion graphs correlating total length of all dendrites morphological parameters with peak amplitude electrophysiological parameters, existing positive correlation with VC recordings in pulses 1 (**A**) and 2 (**B**) and in CC recordings in pulses 1 (**C**) and 2 (**D**) (simple linear regression). Dots represent mean, horizontal graphs morphological (X-axis) SD, and vertical graphs electrophysiological (Y-axis) SD.

**Figure 4 pharmaceutics-15-01860-f004:**
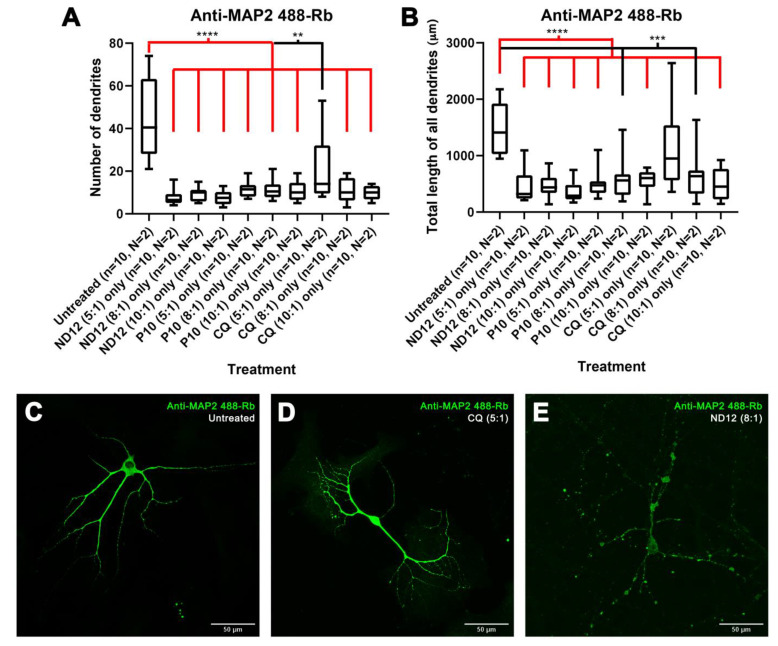
**Treatment with only niosomes in 21-28 DIV rat cortical neurons.** (**A**,**B**) The 21–28 DIV rat cortical neurons treated with only niosomes showed morphological alterations in their number of dendrites (**A**) and total length of all dendrites (**B**) compared with untreated neurons, except for the neurons treated with only CQ niosomes at a 5:1 proportion (Mann–Whitney test, ** *p* < 0.01, *** *p* < 0.001, **** *p* < 0.0001, n = number of cells, N = number of cultures). (**C**,**E**) Morphological aspect of cortical neurons untreated (**C**), treated with only CQ niosomes at a 5:1 proportion (**D**), and only ND12 niosomes at an 8:1 proportion (scale bar = 50 µm) (**E**).

**Figure 5 pharmaceutics-15-01860-f005:**
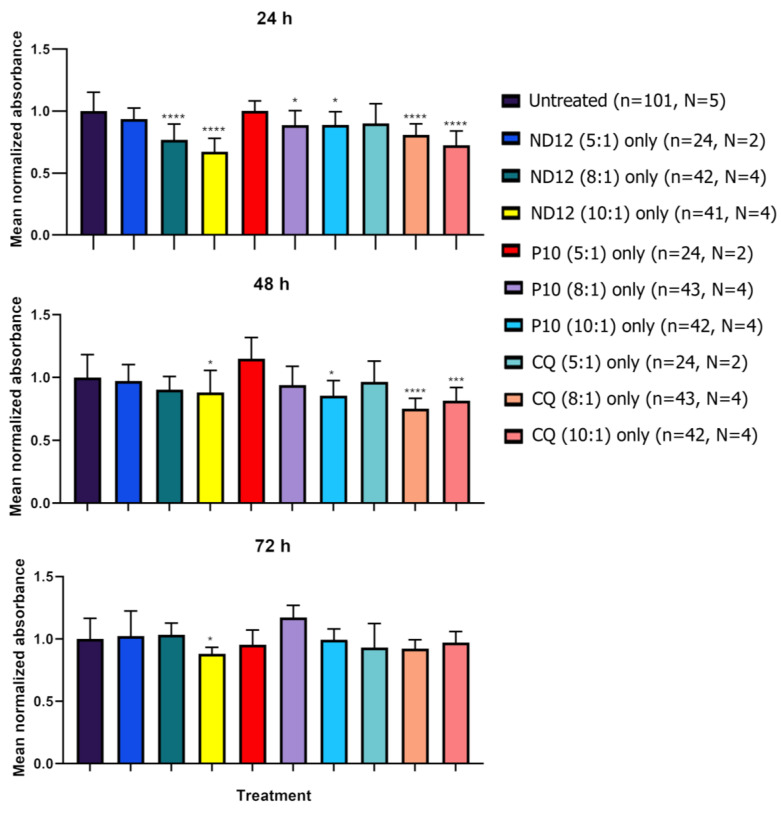
**Cell viability using different proportions of niosomes in 21–28 DIV cortical neurons.** MTT assays performed at 24, 48, and 72 h in 21–28 DIV rat cortical neurons showed reduced cell viability at niosome 8:1 and 10:1 proportions compared with lower proportions and untreated neurons (Multiple *t*-tests, * *p* < 0.05, *** *p* < 0.001, **** *p* < 0.0001, n = number of wells, N = number of cultures). Graph bars are expressed as mean ± SD.

**Figure 6 pharmaceutics-15-01860-f006:**
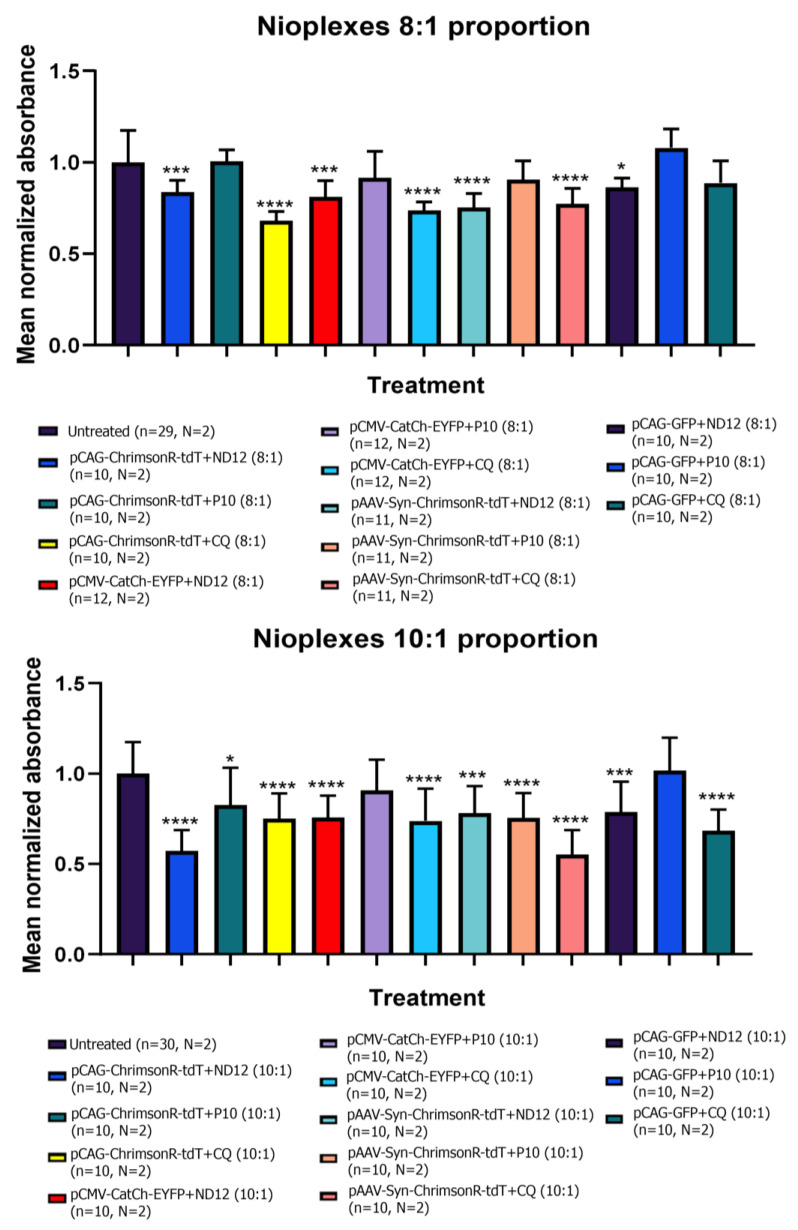
**Cell viability using different proportions of nioplexes.** MTT assays performed after 24 h of treatment in 21–28 DIV rat cortical neurons showed reduced cell viability with nioplexes both at 8:1 and 10:1 proportions compared with untreated neurons (Multiple *t*-tests, * *p* < 0.05, *** *p* < 0.001, **** *p* < 0.0001, n = number of wells, N = number of cultures). Graph bars are expressed as mean ± SD.

**Figure 7 pharmaceutics-15-01860-f007:**
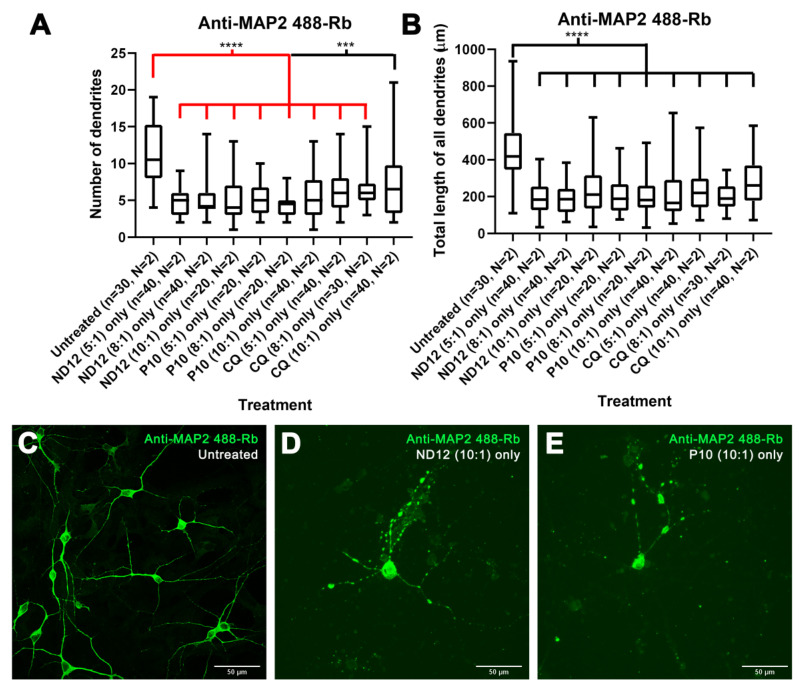
**Morphological changes induced by different niosome-based formulations in 7–11 DIV rat cortical neurons.** (**A**,**B**) The 7–11 DIV rat cortical neurons treated with only niosomes showed morphological alterations in their number of dendrites (**A**) and total length of all dendrites (**B**) compared with untreated neurons (Mann–Whitney test, *** *p* < 0.001, **** *p* < 0.0001, n = number of cells, N = number of cultures). (**C**–**E**) Morphological aspect of cortical neurons untreated (**C**), treated with only ND12 niosomes at a 10:1 proportion (**D**), and only P10 niosomes at a 10:1 proportion (scale bar = 50 µ) (**E**).

**Figure 8 pharmaceutics-15-01860-f008:**
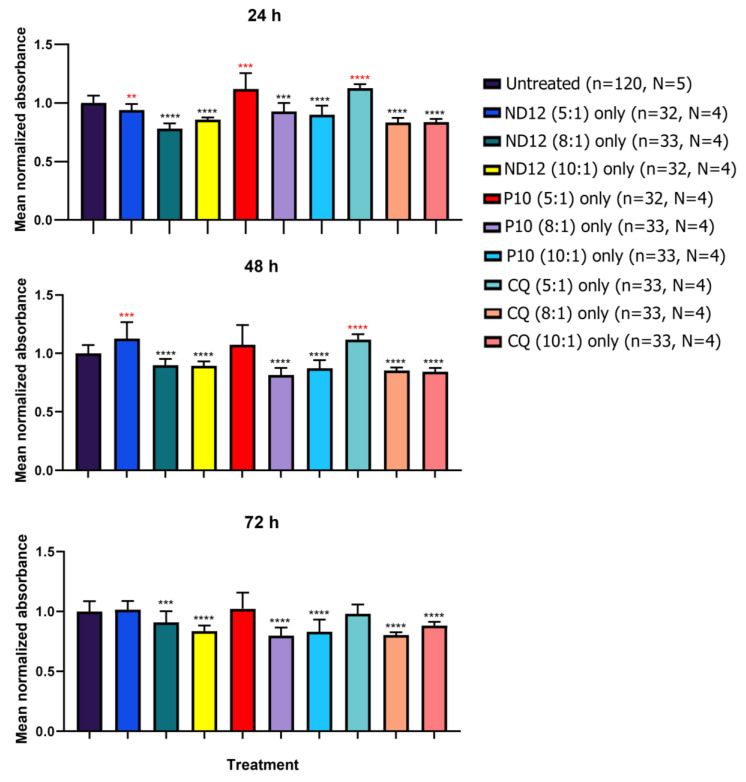
**Cell viability using different proportions of niosomes in 7–11 DIV cortical neurons.** MTT assays performed at 24, 48, and 72 h in 7–11 DIV rat cortical neurons showed reduced cell viability at niosome 8:1 and 10:1 proportions compared with lower proportions, naked plasmid treatments, and untreated neurons (Multiple *t*-tests, ** *p* < 0.01, *** *p* < 0.001, **** *p* < 0.0001, n = number of wells, N = number of cultures). Red *p*-values mean that there is a statistical difference with groups with higher mean values than the untreated groups, while black *p*-values mean that there is a statistical difference with groups with lower mean values than the untreated groups. Graph bars are expressed as mean ± SD.

## Data Availability

Data are contained within the article and the Appendix A.

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
