# Peer review of "Assessment of Different Niosome Formulations for Optogenetic Applications: Morphological and Electrophysiological Effects"

_pharmaceutics, 2023, doi:10.3390/pharmaceutics15071860_

Round 1

Reviewer 1 Report

This work here compared the efficacy of three niosome formulations carrying plasmids expressing optogenetic protein in rat cortical neurons. The varying changes in the dendritic morphology, electrophysiological properties and cell viability of the transfected cells were mainly taken into consideration. Although the conclusion are well supported by the above results, some questions are still necessary to be solved until the final publication. I provided some following points for the consideration.

Major points:

1) The characterization of mRNA expression, protein expression, and fluorescence intensity of three niosome formulations in 7-11 DIV rat cortical neuron cells should be performed to systemically compare their actual efficacy in plasmid delivery and gene expression.  

2) What about the serum stability of three niosome formulations?

3) Are all these formulations at different ratios charaterized by gel assay and TEM? After the preparation of nioplexes at different cationic lipid:DNA ratios, are these nioplexes further purified? 

4) What about the encapsulation efficiency of three niosome formulations? I am worried about the incomplete encapsulation, which will evoke the cytotoxicity, which has been found in the cell viability assay. In this context, the purification will be a critical step.

Minor points:

1) The statistical mark P and p could be uniformed.

2) Some typos need further check. For example, in table S7, -23± 0,3, 0,35±0.07.

Reviewer 2 Report

After careful reading of the manuscript entitled “ Assessment of different niosome formulations for optogenetic 2 applications: morphological and electrophysiological effects” I have the following comments:

1.     Section 2.4, “Afterwards, chemical dissociation was carried out in FBS-free DMEM adding trypsin 0.05% and incubating the mixture at 37 ºC.” this part is unclear. Please revise. Also, Line 149 the word medium is repeated “…..Optimized Medium (BPI; STEMCELL®) medium supplemented with….”

2.     Section 2.5, please correct the word “temperatura”

3.     Section 2.5, Why authors used Neurobasal culture medium in cells transfected with nioplexes and BPI medium in cells transfected with naked plasmid or only niosomes? And why they did not test the 3 conditions (nioplexes, noisome and plasmid) in the same cell line with the same culture medium?

4.     Section 2.5, Could you explain this phrase please “Baseline cells incubated with OptiMEM for 4 h were used as a positive control.”

5.     Section 2.6; please indicate the used concentration of anti-rabbit MAP2 monoclonal antibody and correct the PB 0,1 M to PB 0.1 M.

6.     Please indicate in the methodology the cell viability for cells transfected with the nioplexes  as you only mentioned naked plasmid and noisome.

7.     Authors indicated in section 2.5 “Cells in Neurobasal™ medium were seeded and incubated between 21-28 days in  vitro (DIV) before transfecting” then in section 3.1 they indicated “More specifically, cortical neurons were transfected with the niosomes ………….. at different cationic lipid/genetic material ratios (2:1, 5:1, 8:1 and 10:1) between 21 and 28 days in vitro (DIV).” section 3.2 “cells transfected after 21-28 DIV with our niosome complexes” which phrase is correct? Did you incubate the cells for 21-28 days before transfection or this period is the transfection period?

8.     In the methodology authors used 3 terms refereeing to the 3 conditions (nioplexes, noisome and plasmid) but in the results all are mixed, especially in section 3.1 Nioplexes transfection, I found this phrase “However, the morphology of niosome transfected cells appeared somehow different in comparison to cells treated with the commercially available reagent Lipofectamine.” I suggest the authors to homogenize the text.

9.     In section 3.1, authors reported “ To quantify and characterize the differences, we measured: (1) the number of dendrites, (2) branching points, (3) …..” you are missing the (5) the longest Longest. Please add.

10.  Please mention Figure 2B to 2F in the text.

11.  Section 3.2, line 293, “and P10 (10:1) treatment in pulse 2 (p-value= 0.2445 in VC) (Supplementary Fig. S4F) » please revise as from the Figure it seems pulse 1 not pulse 2.

12.  Section 3.2, line 301, “and CQ (10:1) treatment in pulse 1 (p-value = 0.4276)” please add Pulse 1 for CQ (8:1)”

13.  In the methodology section 2.7, authors reported “ A monochromatic light source …………… and 470 nm for CatCh positive cells” in the legend of Figure S5, authors indicated 590 nm as the wavelength. Please check.

14.  Why authors performed experiments on 7-11 DIV cells only for naked plasmid and only niosomes and not for the nioplexes?

15.  Why authors did not comment the results of pCAG-ChrimsonR-tdT naked of Figure S8?

16.  The title of section 3.3 “Niosome formulations, neuron morphology and viability.” Should be modifying and describing the results that deal with naked plasmid and niosomes only.

17.  Line 385, “ At ratios 8:1 and 10:1 we observed significant decreases in cell viability for all plasmids tested” please replace the plasmids with nioplexes

18.  I wonder, visually from the Figure 5, ND12 (8:1) after 24 h is around 0.75 and when we see the values in Figure 6 for the nioplex at the same ratio with all the plasmids the viability is around (CAG 0.9, Catch 0.8 and Syn 0.75) and I notice that for all the neosomes and nioplexes values. so how you said that the viability values of niosomes are higher than nioplexes.

19.  For easy read and visualization of the cell viability, I suggest the authors to  put in the same curve one neosome (with its different ratio) with its nioplexes.  

20.  Lines 385-387 “ At ratios 8:1 and 10:1 we observed significant decreases in cell viability for all plasmids tested (Fig.6) except for P10 at 10:1 for all combinations except for when complexed with CAG- ChrimsonR (p-value=0.0129*) .” Please revise.

21.  Line 388 “GFP nioplexes also resulted in decreased viability for all ratios and niosome combinations except for CQ 8:1 (p-value=0.7845) (Fig. 6).” What about P10?

22.  Line 389, “When we only applied the plasmids or Lipofectamine we observed no decrease in cell viability.” Please mention the Figure S9.

23.  In line 410, authors reported “ These results suggest that older 410 neurons appear to be more resistant to niosome induced alterations” in term of toxicity the 7-11 DIV seems more resistant to neosomes than older ones. Could you explain more?

24.  Could the authors provide the DLS and zeta potential for the neoplexes?

25.  Could the authors provide the concentrations of the neosomes and plamids at each ratio?

Reviewer 3 Report

pharmaceutics-2375563

Assessment of different niosome formulations for optogenetic applications: morphological and electrophysiological effects

The manuscript by Celdrán et al. described the development of different niosome formulations to load optogenetic plasmids in rat cortical neurons. The authors demonstrated that all the developed niosomes were able to transfect cortical neurons with optogenetic channels. They also found changes in the dendritic morphology and electrophysiological properties of transfected cells, which posed a critical issue to consider. The authors provided sufficient data for conclusions. However, the manuscript should be improved considering some comments below.

1. Abstract and Introduction: the purpose of this study is unclear. The authors should clarify the reasons for delivering optogenetic plasmids ChrimsonR and CatCh, what they are used for, and which delivery systems have been used previously to deliver them. From those, the authors should highlight the necessity of developing plasmid-loaded niosomes.

2. It is better to include a Material section.

3. Lines 51-53: the authors should avoid citing many citations at once, particularly self-citation.

4. Line 124: is it correct (“Д)?

5. Please check and correct the decimal separator throughout the manuscript and supplementary files. There are some errors, such as line 276 and table S7.

6. Please use “×” instead of “*” for multiplication symbols.

Moderate editing of the English language is required

Round 2

Reviewer 1 Report

The manuscript has been well revised. All my concerns are answered. It can be accepted. This work will be stimulating the futher thinking. 

Author Response

Thanks a lot for all your help and advice.

Reviewer 2 Report

I would like to thank the authors for their effort to improve the manuscript. I still have few comments:

1-    Section 2.4, line 150, the word “by” is missing in the following phrase “Chemical dissociation was carried out by adding trypsin 0.05%”.

2-    What the authors mean by the letter “p” in 24P or 96P well plates?

3-    Section 2.5, line 177, please remove “in each well” as I guess your incubation for 30 min was carried out on Eppendorf on the bench followed by transfection of cells (in plate) for 4 h. Further, please take a space in “Untreatedcells”

4-    Could you please add your answer for the comments N° 3 and 14 to the manuscript?  

5-     Concerning the answer to comment N° 15. Please add also the P value of the total length of all dendrites.

6-    Along the whole manuscript, it is still confusing the terms that the authors used.  I guess niosome formulations are niosome alone, a term that you used too. If I am right, please choosing 3 precise terms for your conditions and uniform your text for easy read and please pay attention to the Figure legends too.  

7-    Line 406. please put a point between GFP and (p-value = 0.1531).

8-    I do appreciate that the authors provided the characterization of the nioplexes, I see that the new measurement values of the niosome alone are different than before and I can understand, but I cannot understand how ND12 moves from negative charge to positive charge in the new measurements?  Do you have an explanation?
